

# Can we set a global threshold age to define mature forests?

Philip Martin[1], Martin Jung[2], Francis Q. Brearley[3], Relena R. Ribbons[4,5], Emily R. Lines[6] and Aerin L. Jacob[7]

[1] Centre for Conservation Ecology and Environmental Science, Bournemouth University, Bournemouth, United Kingdom
[2] School of Life Science, University of Sussex, Brighton, United Kingdom
[3] School of Science and the Environment, Manchester Metropolitan University, Manchester, United Kingdom
[4] School of Environment, Natural Resources and Geography, Bangor University, Bangor, United Kingdom
[5] Department of Geosciences and Natural Resource Management, University of Copenhagen, Frederiksberg, Denmark
[6] School of Geography, Queen Mary University of London, London, United Kingdom
[7] School of Environmental Studies, University of Victoria, Victoria, Canada

Corresponding author
Philip Martin,
phil.martin.research@gmail.com

## ABSTRACT

Globally, mature forests appear to be increasing in biomass density (BD). There is disagreement whether these increases are the result of increases in atmospheric $CO_2$ concentrations or a legacy effect of previous land-use. Recently, it was suggested that a threshold of 450 years should be used to define mature forests and that many forests increasing in BD may be younger than this. However, the study making these suggestions failed to account for the interactions between forest age and climate. Here we revisit the issue to identify: (1) how climate and forest age control global forest BD and (2) whether we can set a threshold age for mature forests. Using data from previously published studies we modelled the impacts of forest age and climate on BD using linear mixed effects models. We examined the potential biases in the dataset by comparing how representative it was of global mature forests in terms of its distribution, the climate space it occupied, and the ages of the forests used. BD increased with forest age, mean annual temperature and annual precipitation. Importantly, the effect of forest age increased with increasing temperature, but the effect of precipitation decreased with increasing temperatures. The dataset was biased towards northern hemisphere forests in relatively dry, cold climates. The dataset was also clearly biased towards forests <250 years of age. Our analysis suggests that there is not a single threshold age for forest maturity. Since climate interacts with forest age to determine BD, a threshold age at which they reach equilibrium can only be determined locally. We caution against using BD as the only determinant of forest maturity since this ignores forest biodiversity and tree size structure which may take longer to recover. Future research should address the utility and cost-effectiveness of different methods for determining whether forests should be classified as mature.

## INTRODUCTION

Forests play an important role in the global climate system, covering nearly one-third of the earth's terrestrial surface and accounting for over three-quarters of terrestrial gross primary production (*Pan et al., 2013*). Forests also provide vital habitats for biodiversity and supply a wide-range of ecosystem services upon which humans depend, such as climate regulation via carbon storage in tree biomass (*Foley et al., 2007*). Globally, mature or old-growth forests, here defined as those that have developed the structures and species associated with old primary forest (*CBD, 2016*), appear to be increasing in biomass density (hereafter referred to as BD) (*Pan et al., 2011*). Mature tropical forests, in particular, have increased in BD by around 0.5 Mg C ha$^{-1}$ year$^{-1}$ (*Baker et al., 2004*; *Lewis et al., 2009*), though the rate of increase now appears to be slowing (*Brienen et al., 2015*).

Some researchers have hypothesised that increased $CO_2$ concentrations in the atmosphere, as a result of human activities, have stimulated the growth of trees in mature forests, resulting in increased BD (*Lewis et al., 2009*). However, other researchers reject these claims, hypothesising that that many mature forests are in fact undergoing secondary succession following 'unseen' disturbances that occurred either hundreds of years ago (*Brncic et al., 2007*; *Muller-Landau, 2009*) or as a result of extreme weather such as El Niño events (*Wright, 2005*). If many supposedly mature forests are recovering from previous human influence, then this may account for observed increases in BD (*Wright, 2005*). It is thus unclear whether the mature forests in studies that showed increases in BD were actually old enough to achieve a state of relative equilibrium, which can take decades to centuries. In addition, mature forests are commonly used as a reference against which to compare biodiversity in degraded forests or alternative land-uses. If a large proportion of mature forests are actually recovering from disturbances themselves, this may lead to an overestimation of the conservation value of degraded forests. However, until recently, there has been no attempt to determine whether there are methods that could be applied globally to enable forests recovering from disturbances to be distinguished from relatively stable mature forests.

Recently, *Liu et al. (2014)* attempted to address this issue by (i) assessing how climate and forest age affect forest BD, and (ii) using this analysis to define an age threshold for mature forests. The authors concluded that the BD of mature forest stands was highest in areas with a mean annual temperature of *c*. 8–10 °C and mean annual precipitation between 1,000 and 2,500 mm (*Liu et al., 2014*). In addition, the authors further suggested that forest BD increased with stand age, plateauing at approximately 450 years of age (*Liu et al., 2014*). However, given that previous work has shown that climate strongly influences both biomass accumulation (*Johnson, Zarin & Johnson, 2000*; *Anderson et al., 2006*; *Anderson-Teixeira et al., 2013*) and the maximum BD attainable by a forest (*Stegen et al., 2011*) it seems unlikely that there is a single global *age* threshold that can be used to define mature forests. Rather if such thresholds are used, they will need to be defined in areas with relatively homogenous climates where accumulation rates and maximum attainable BD vary relatively little.

To address these issues we use a subset of the data used by *Liu et al. (2014)* to revisit the questions:

1. How do climate and forest age control the biomass density of global forests?
2. Can we use this to set an age threshold for mature forests globally?

While the analyses we present here use the same data as *Liu et al. (2014)*, we improve on their analyses by considering interactions between precipitation, temperature and estimated forest age. Our work shows that these interactions improve model fit considerably, as well as indicating that establishment of a single age threshold for mature forests is ecologically unrealistic.

## METHODS

The data we used for this study were taken from *Liu et al. (2014)* in which the authors tested global-scale correlations between mature forest carbon stocks (biomass density), stand age and climatic variables using data collected from previous studies. Here we used a subset of these data on aboveground biomass density (AGB, Mg ha$^{-1}$) for sites that had estimates of forest age (years), mean annual precipitation (mm), mean annual temperature (° C) and geographic location (latitude and longitude). Details of the studies from which data was taken can be found in Table 1.

To examine our first question of how forest BD is determined by climate and forest age we used linear mixed effect models (LMMs). First, we tested whether accounting for methodological differences between studies and spatial autocorrelation improved model performance compared to null models. To do this, we fitted a model with a dummy random effect and compared the corrected Akaike Information Criteria (AICc) value to our null models, which included study level random effects and a matrix to account for spatial autocorrelation. Using the random effects structure deemed most parsimonious, we then tested the effects of temperature, precipitation and forest age on AGB by running all possible LMMs that included two way interactions as well as less complex additive models. Forest age was log transformed as increases in AGB with age tend to be non-linear (*Martin, Newton & Bullock, 2013*). All explanatory variables were standardised following *Schielzeth (2010)*, by subtracting the mean from each value and dividing by the standard deviation. This method allows easier interpretation of coefficients and improves model convergence. To reduce heteroscedasticity in model residuals we log transformed the response variable. Conditional $R^2$ values were calculated using the methods of *Nakagawa & Schielzeth (2013)*.

Models were ranked by AICc and model averaging performed using all models with a $\Delta$AICc $\leq 7$ to produce coefficient estimates (*Burnham & Anderson, 2002*; *Burnham, Anderson & Huyvaert, 2010*). These coefficient estimates were subsequently used to predict AGB in relation to stand age, precipitation and temperature. Based on our results we then inferred an answer to our second question, relating to age thresholds of forest maturity. Variable importance values were calculated for each coefficient and interaction as recommended by *Burnham & Anderson (2002)* by combining the AICc weight of each model in which the variable was included. Calculating the importance

values allows the relative importance of each variable in explaining the relationship to be determined (*Burnham & Anderson, 2002*). If interactions between climate and forest age were considered to be relatively important we determined that it was not possible to set a global age threshold by which to define mature forests without considering their local characteristics.

It is important in analyses that combine data from multiple sources to determine whether the data being used show signs of bias that might influence a study's results. One example of such a bias is if data is not representative of an overall population which it seeks to represent (*Tuck et al., 2014*). In the case of our study, such bias may be caused by an over- or under-representation of particular forest ages, certain climates and particular geographic regions. To test for this we (i) examined the age distribution of forests using histograms; (ii) determined the climate space encompassed by the sites used in this study compared to that occupied by forests globally; (iii) and examined the geographical distribution of study sites. For the comparison of the forest climate space we binned the data on precipitation into bins of 200 mm and mean annual temperature into bins of 1 °C. We then used a global grid with a resolution of 0.5 decimal degrees to identify where forest was present based on the globcover 2009 dataset (*Bontemps et al., 2011*). We determined the mean total precipitation and mean annual temperature in each grid cell where forest was present using WorldClim (*Hijmans et al., 2005*). We then compared the percentage of our data contained within each temperature and precipitation bin with the percentage area of global forests contained in each bin. All analyses were conducted in R version 3.2.1 (*R Development Core Team, 2008*) and with models produced using the nlme (*Pinheiro et al., 2015*) and MuMIn packages (*Barton, 2015*). All R scripts used for analyses can be found at https://github.com/PhilAMartin/Liu_reanalysis.

## RESULTS

Our model-averaged results indicated positive relationships between AGB and the logarithm of forest age (slope = 0.24, SE = 0.02, Importance value = 1), mean annual temperature (slope = 0.18, SE = 0.04, Importance value = 1) and mean annual precipitation (slope = 0.32, SE = 0.04, Importance value = 1). Importantly, the slope related to forest age increased with mean annual temperature and was considered to be relatively important (interaction term = 0.06, SE = 0.02, Importance value = 0.85). In addition, the positive effect of mean annual precipitation on AGB was reversed at higher temperatures (interaction term = −0.12, SE = 0.02, Importance value = 1). The interaction term between precipitation and forest age was considered to be less important in characterising forest BD (interaction term = −0.02, SE = 0.02, Importance value = 0.34). Models included in the model averaging process had reasonable descriptive power with conditional $R^2$ values varying from 0.18 to 0.24. Predictions using model-averaged coefficients did not show a clear plateauing of AGB at any age (Fig. 1), contrary to the findings of *Liu et al. (2014)*. These models also had greater support than the models of *Liu et al. (2014)*, containing only age, precipitation and temperature ($\Delta$AICc = 112.41, 114.17 and 139.99 respectively), which also showed poor descriptive power (Conditional $R^2$ = 0.03, 0.04 and 0.02 respectively, Table 2).

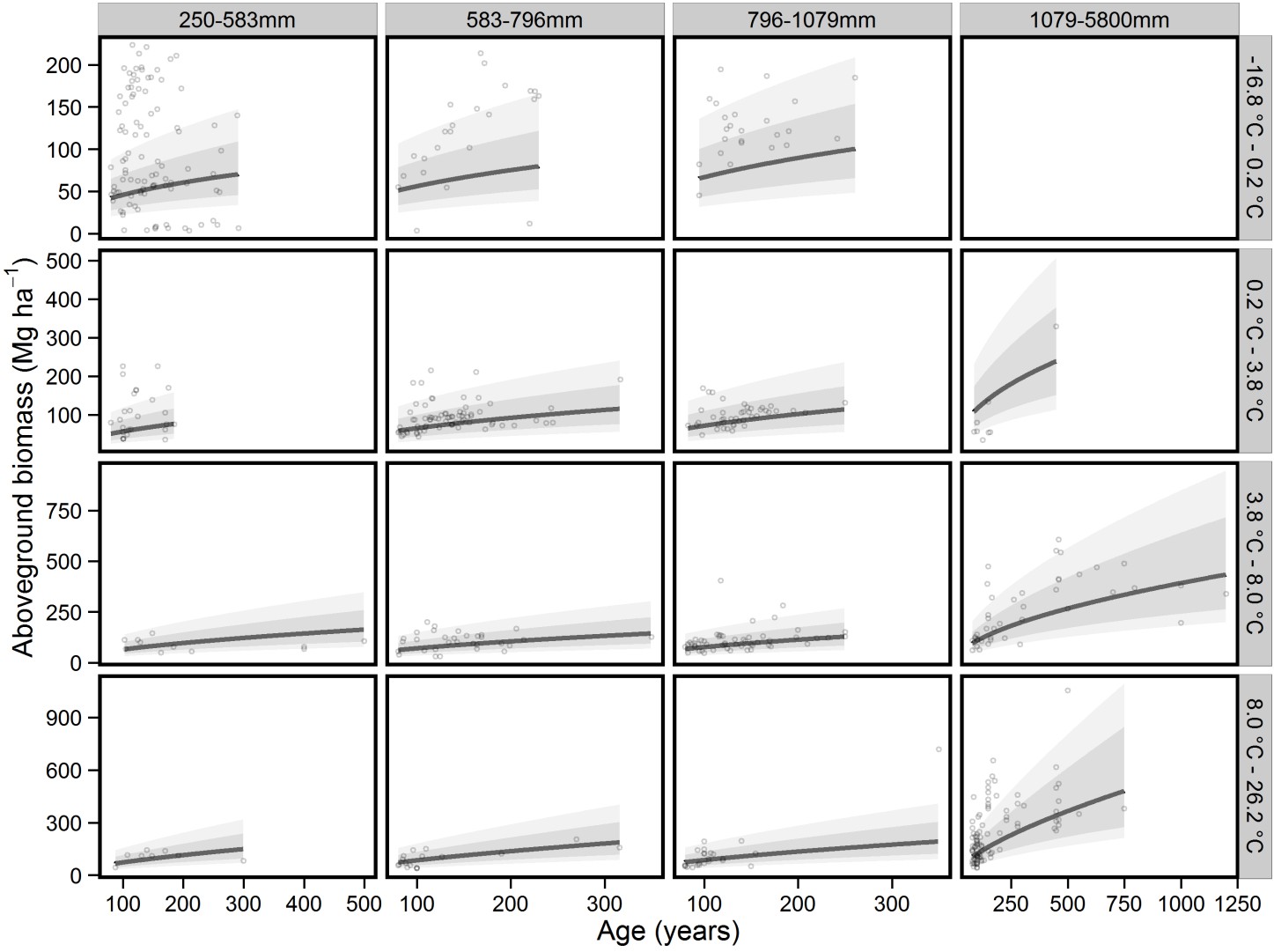

**Figure 1** **The relationship between forest age and aboveground biomass for differing climate spaces.** Panels represent binned mean annual temperature (rows) and total annual precipitation (columns). Points represent individual sites and solid lines predictions from model-averaged coefficients of models with a $\Delta$AICc $\leq$ 7. The dark band around predictions represents the 95% confidence interval of the coefficient, with the lighter band representing the 95% confidence interval when uncertainty in random effects is taken into account. Bins represent quartiles for precipitation and temperature so that each bin contains a similar number of data points. Please note that the $y$ axes are not the same for all panels.

There are clear biases in the dataset we used for this analysis. Tropical and southern hemisphere forests are under-represented, relative to the area they cover (Fig. 2A). While the data we used also covered a wide range of climatic conditions, there was a bias towards forests found in relatively cold, dry climates and away from warmer, wetter climates (Fig. 2B). The dataset we used was also clearly biased towards younger forests, containing relatively few stands >250 years of age (Fig. 2C); although we note that the ages of many tropical sites appear to be an uncritical reference to *Luyssaert et al. (2007)*, where the ages of the stands in a range of minimally disturbed tropical forests was reported as being between 100–165 years old.

**Table 1   Characteristics of studies used in this paper.**

| Reference | Mean annual temperature ( °C) | Mean annual precipitation (mm) | Mean forest age (years) |
|---|---|---|---|
| Bondarev (1997) | −13.3 | 290 | 190 |
| Liu et al. (2011) | 13.6 | 1,235 | 87 |
| Chang et al. (1997) | −3.7 | 347 | 204 |
| China's Forest Editorial Committee (1999) | −1.0 | 470 | 216 |
| Feng, Wang & Wu (1999) | 9.0 | 850 | 350 |
| Hudiburg et al. (2009) | 7.8 | 2,276 | 423 |
| Kajimoto et al. (2006) | −9.8 | 610 | 158 |
| Keeton et al. (2010) | 7.0 | 800 | 217 |
| Keith, Mackey & Lindenmayer (2009) | 10.7 | 1,593 | 500 |
| Liu et al. (2014) | −3.2 | 596 | 163 |
| Luo (1996) | 5.2 | 889 | 130 |
| Luyssaert et al. (2007) | 7.3 | 1,204 | 162 |
| Ma et al. (2012) | −0.1 | 618 | 137 |
| Tan et al. (2011) | 11.3 | 1,840 | 300 |
| Zhou et al. (2002) | −4.7 | 446 | 149 |
| Zhu et al. (2005) | −2.0 | 459 | 84 |

**Table 2   Candidate mixed effect models for explaining global forest biomass density.**

| Formula | Model rank | df | log likelihood | AICc | ΔAICc | AICc weight | Conditional $R^2$ |
|---|---|---|---|---|---|---|---|
| A + T + P + A*T + T*P | 1 | 10 | −305.02 | 630.44 | 0 | 0.56 | 0.25 |
| A+ T+ P+ A*T+ T*P+ A*P | 2 | 11 | −304.61 | 631.70 | 1.26 | 0.3 | 0.25 |
| A + T + P + T*P | 3 | 9 | −307.74 | 633.81 | 3.37 | 0.1 | 0.21 |
| A + T + P + T*P | 4 | 10 | −307.74 | 635.88 | 5.44 | 0.04 | 0.21 |
| A + T + P + A*T | 5 | 9 | −318.73 | 655.79 | 25.35 | <0.01 | 0.15 |
| A + T + P + A*T + A*P | 6 | 10 | −318.43 | 657.25 | 26.82 | <0.01 | 0.16 |
| A + T + P + A*P | 7 | 9 | −319.98 | 658.28 | 27.85 | <0.01 | 0.12 |
| A + T + P + A | 8 | 8 | −321.03 | 658.32 | 27.88 | <0.01 | 0.14 |
| A + P | 9 | 7 | −329.94 | 674.08 | 43.64 | <0.01 | 0.10 |
| A + P + A*P | 10 | 8 | −329.74 | 675.73 | 45.3 | <0.01 | 0.10 |
| A + T + A*T | 11 | 8 | −333.58 | 683.42 | 52.98 | <0.01 | 0.11 |
| A + T | 12 | 7 | −335.71 | 685.63 | 55.19 | <0.01 | 0.09 |
| T + P + T*P | 13 | 8 | −350.23 | 716.72 | 86.28 | <0.01 | 0.12 |
| T + P | 14 | 7 | −363.42 | 741.04 | 110.6 | <0.01 | 0.04 |
| A | 15 | 6 | −365.35 | 742.84 | 112.41 | <0.01 | 0.03 |
| P | 16 | 6 | −366.23 | 744.61 | 114.17 | <0.01 | 0.04 |
| T | 17 | 6 | −379.14 | 770.43 | 139.99 | <0.01 | 0.02 |
| Null model | 18 | 5 | −395.95 | 802.01 | 171.57 | <0.01 | 0.00 |

**Notes.**

A, Age; T, Temperature; P, Precipitation.
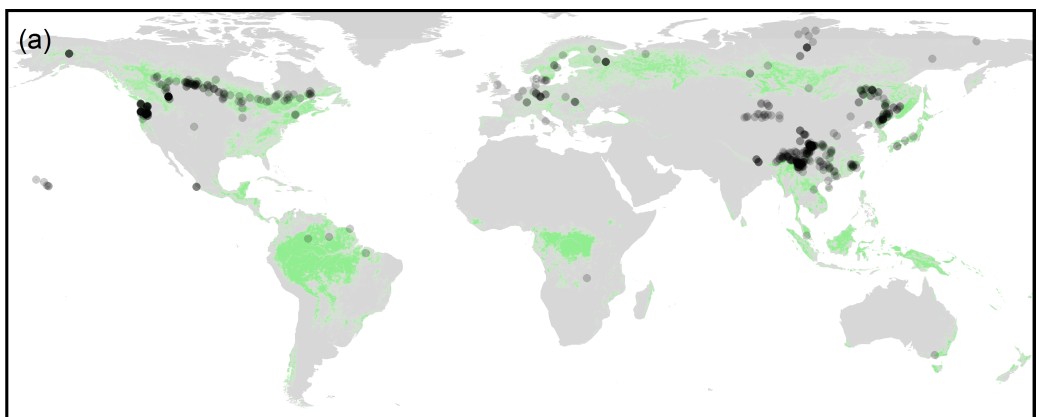

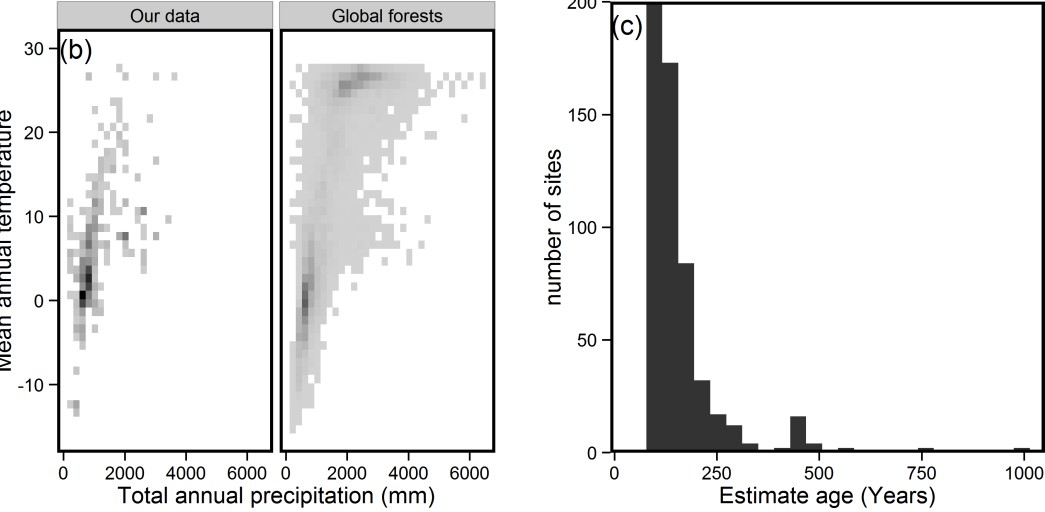

**Figure 2   Potential biases associated with the dataset we used for this study.** (A) Spatial distribution of sites used in this study, showing lack of tropical sites. Green areas represent forest, dots sites used in this study. Dots are partially transparent to give an impression of site density. (B) Climate space represented by data used in this study and forests globally (climate data from *Hijmans et al. (2005)*, forest cover data from *Bontemps et al. (2011)*). Darker pixel colour indicates greater density of data, indicating a bias towards forests with low precipitation and low mean annual temperature. (C) Distribution of sites used in this study by site age, showing a bias towards forests <250 years old.

## DISCUSSION

Our results indicate that climate and forest age interact to determine aboveground biomass density in global mature forests. This study is, to our knowledge, the first to quantitatively show this interaction. Previous studies have shown that biomass accumulation rate of regrowing forests depends on precipitation and temperature (*Johnson, Zarin & Johnson, 2000*; *Anderson et al., 2006*; *Anderson-Teixeira et al., 2013*) and that climate is an important constraint of BD in mature forests (*Stegen et al., 2011*). We show that forests experiencing higher temperatures accumulated biomass more rapidly, in agreement with previous studies (*Anderson-Teixeira et al., 2013*). However, our results

also suggest that there is little interaction between forest age and annual precipitation. Taken together, these results support the findings of *Anderson et al. (2006)* that, on a global scale, temperature drives differential rates of biomass accumulation. However, reality is likely to be more complex than our results suggest. For example, *Stegen et al. (2011)* suggested that water deficits resulting from interactions between precipitation and temperature are a primary limiting factor of the BD that can be attained by mature forests.

In contrast to the recent study of *Liu et al. (2014)*, we found that it is not possible to set a threshold age at which forests can be considered mature. Since our results indicated that aboveground BD was best determined by models that included interactions between climate and stand age, any threshold age for mature forests must be determined at a relatively local scale. Accumulation of biomass varies locally with soil nutrient content and drainage, distance to other forest patches and previous land-use (*Norden et al., 2015*). In addition, local effects such as priority effects, herbivore density, invasive species, pathogen presence and edge effects can all result in unpredictable successional pathways (*Norden et al., 2015*). Setting a biome-level threshold age at which forests could be considered mature as *Liu et al. (2014)* did in part of their paper may provide a compromise between global and local thresholds. However, any such estimates must be explicit about the variation in forest trajectories across a given biome.

## THE NEED FOR BETTER DATA

Though our analysis is an improvement on that performed by *Liu et al. (2014)*, we were limited by the representativeness of the data used. These data comprised few tropical forest sites, were biased towards relatively cold, dry forests and very few forests >250 years of age were included in the dataset. The lack of data from tropical forests limits the generality of this analysis meaning that we have little confidence about extrapolating our results to the tropics. This is particularly important as tropical forests store approximately one third of global terrestrial carbon (*Dixon et al., 1994*) and appear to be increasing in BD (*Baker et al., 2004*; *Lewis et al., 2009*). As such, our analysis and that of *Liu et al. (2014)* can say nothing about whether the recent increases in BD in apparently mature tropical forests may be the result of recovery from past disturbances as *Liu et al. (2014)* suggested. The relative lack of data for forests >250 years of age in our study limits our conclusions, given that forests are often thought to take 100–400 years to reach maturity (*Guariguata & Ostertag, 2001*).

Critically, the setting of any threshold requires accurate aging of forests. This is not a trivial task. In mature forests trees are recruited as other die, creating a complex patchwork of differently aged trees (*Chazdon, 2014*). As such, defining the age of a forest using the oldest tree (as studies that we used data from did) will likely only be accurate in relatively young forests where tree ages do not differ greatly. However, in mature forests where all pioneer individuals have been replaced, the age of the oldest tree no longer provides a useful determinant of forest age. Thus, the precision of our estimates for younger forest are undoubtedly greater, and more useful, than for older forests. Furthermore, as most tropical trees lack annual growth rings, $^{14}$C dating is the only feasible way to currently age most tropical trees and this is prohibitively expensive in most cases.

## DIFFICULTIES OF DEFINING MATURE FORESTS

While, in the future, it may be possible to determine at what age forest BD becomes relatively stable, we advise against using this as a definition of forest maturity for three reasons. Firstly, while carbon storage in the form of BD is important from the perspective of alleviating the impacts of climate change, it tends to recover relatively quickly. In tropical secondary forests, community composition of tree species can take >150 years to recover, with BD recovering in approximately 100 years (*Martin, Newton & Bullock, 2013*). Thus, while biomass accumulation is important, using it alone to define forest successional stage may lead to forests being classified as mature, when they are still undergoing the latter stages of succession. Incorrect classification of forests as mature based solely on aboveground may mean that when comparisons of biodiversity are made with degraded forests, the conservation value of these degraded forests is overestimated. Secondly, though mature forests can appear to be relatively stable when observed over short time periods or as part of a chronosequence, they never reach equilibrium. Over decadal time scales even apparently mature forests rarely show stable BD (*Valencia et al., 2009*), and are influenced by changes in climate and changes in local land use. Thirdly, one characteristic of old-growth forests is that they do not contain any individual trees that colonised immediately following allogenic disturbances (*Chazdon, 2014*). As such, forests that contain remnant cohorts of long-lived pioneer species should be considered as late successional rather than old-growth forests (*Chazdon, 2014*). Thus, examining changes in biomass is likely to be of little use in separating late successional forests such as these from true old-growth.

The results of this study and others clearly show it is challenging to define whether a forest should be classed as mature. Previous studies of forest succession suggest that biomass density could be used along with size structure of tree populations (*Coomes et al., 2003*; *Coomes & Allen, 2007*) and species composition to determine maturity (*Chazdon, 2014*). For example, work in temperate forests has shown that during succession the diameter distribution of trees become more symmetrical due to lower recruitment rates under closed canopies (*Coomes & Allen, 2007*). Examination of a forest's diameter distribution, and especially how it changes over time, can also help to identify issues relating to recruitment limitation or high mortality of large trees (*Ghazoul et al., 2015*; *Martin et al., 2015*). Similarly, the size distribution of forest gaps measured with airborne Lidar can be used to distinguish between mature forests and those that have previously been logged and are now undergoing recovery (*Wedeux & Coomes, 2015*; *Kent et al., 2015*). The presence of species characteristic of old-growth conditions, such as trees tolerant of deep shade, may also aid classification of mature forest. Methods such as those developed by *Chazdon et al. (2011)* are particularly useful as they allow an objective, statistical method to classify species as generalists, pioneers or old-growth species. Using this method *Chazdon et al. (2011)* showed old-growth and generalist species increased in abundance while pioneer species abundance decreased during succession in a lowland forest in Costa Rica. Ideally, identification of whether a forest should be classified as mature would encompass all of our suggestions, monitored over a number of years. However, relatively few studies

have combined these different methods. Research that identifies the usefulness and cost-effectiveness of different metrics for defining forests as mature would help to improve the baseline against which the world's increasingly degraded forests are compared.

### Funding
The authors received no funding for this work.

### Competing Interests
The authors declare there are no competing interests.

### Author Contributions
- Philip Martin conceived and designed the experiments, performed the experiments, analyzed the data, wrote the paper, prepared figures and/or tables, reviewed drafts of the paper.
- Martin Jung performed the experiments, analyzed the data, wrote the paper, prepared figures and/or tables, reviewed drafts of the paper.
- Francis Q. Brearley, Relena Ribbons, Emily R. Lines and Aerin L. Jacob wrote the paper, prepared figures and/or tables, reviewed drafts of the paper.

### Data Availability
Figshare: https://figshare.com/articles/Data_for_paper_on_categorising_forest_maturity/2060970.

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
