# Peer review of "Can we set a global threshold age to define mature forests?"

_PeerJ, doi:10.7717/peerj.1595_

## Round 0.1 · original submission · Minor Revisions

Both reviewers offer good suggestions and ask important questions that can help to improve the manuscript if properly addressed.

·

Basic reporting

The article is well structured and well written. It follows a sound logic and it explains clearly what the aims of the project are and why.

Figure 1:
- Please repair ° symbols – they are currently displayed as ~*
- How did you decide to bin the temperatures and precipitations?
- I understand that you wanted to keep the age scale the same for each panel. However, I think that the figure would be much more informative if you “fill” each of the panel with information, by restricting the time scale to your available data. You could still keep the scale the same for the precipitations 250-1079 mm, and only expand the scale for the last column, and if you are worried that people might misinterpret, you could emphasize in the figure caption that the scales are different. As it is, the lines and points are so squashed into the corner of each panel, that we can't really tell much from the figure, apart from the fact that the age data spread is better for areas with higher precipitation. Also, could you show the confidence intervals around the lines?

Figure 2:
- Your data set seems to be a subset of Liu et al. 2014 dataset. Is that correct? Maybe I missed this information in the article, but I couldn't find it in the methods. How many data points did you have? Or is it the case that Liu et al. didn't actually use all the data points that they show in their figure 1?

Experimental design

Method:
Methods:

Mixing of Information-Theoretic (model selection) and frequentist (hypothesis testing) approaches. You have taken the information-theoretic approach to rank the candidate models, and then chose, based on AICc, all models with delat AICc < 7 to perform model averaging. I think that this is an appropriate technique to use in your case. However, I don't understand why you are then reporting p values (Results section, lines 154 to 161). What do these p values mean? What hypothesis are you testing, using your averaged model? What does it mean that a certain interaction term is not significant? Presumably, there was a reason why you chose to “trust” the best models and use them for model averaging, so why do you then still “test” them with p values? I think that once you have opted for the model selection approach, you shouldn't mix it with a hypothesis-testing approach. See for example Anderson and Burnham 2002 “Avoiding pitfalls when using information-theoretic methods” for an explanation why this is not recommended.

L118-119: you tested also models without interactions, if I understand correctly (looking at the table of candidate models). The sentence, as it is, sounds as if you only tested models with interactions.

Results:
Can you describe or give reference for conditional R2 values? Is that the same type of R2 that Liu et al. used?

Delta AICc is not a measure of descriptive power of the model. If all your candidate models are very bad, one of them will still have delta AICc 0, even though its descriptive power might be 0.

Details:

L69: remove one “that”
Table 1: no need to put parentheses around reference in the table.

Validity of the findings

The results support well the conclusions that the authors make.

Additional comments

In general, the article is well written. Here are some further comments on the discussion section that you might want to consider (but these are not problems with the manuscript, please treat them just as suggestions):

Discussion:

- L240-242: What consequences might that have in practice? (e.g. for conservation?) What is the concept of a mature forest useful for? This might be good to mention (even in just 1-2 sentences) in the introduction.
- L243-244: if you mean “stable over time”, then by definition we cannot infer anything about stability when observing something at a single point in time.
- L260-262: you mention that using the forestplots.net data set would substantially strengthen your conclusion – why then did you not use the data set? Or did I misunderstand?
- I the introduction, you set out to ask 2 questions. You answer question 1. Then you go on and say that we cannot use this (i.e. age and climate) to set an age threshold for mature forests in terms of biomass, because a) there are interactions, b) because biomass should not be used to set a mature forest threshold anyway. I think that this is sound reasoning and the discussion is well structured to present this argument. However, the last section, “The future of forest biomass assessment” then seems a little out of context – why would you want to pursue further the relationship between biomass and climate, in the context of studying thresholds of maturity, when you say that it doesn't make sense? I understand of course how it is interesting to study this in general, and for other specific reasons, but perhaps you should mention that. Otherwise the readers might think that you actually do think that we need to understand biomass better, in order to still set a thresholds in terms of climatenand age. Or perhaps I misunderstood your intentions in the discussion, in which case it might be worth reconsidering how the argument is presented.
- What do you think about Liu's thresholds of 200, 500, and 300 for boreal, temperate, and subtropical forests? You mention that thresholds should be determined at a local scale – how large do you think the local scale should be? Is regional (e.g. boreal) not good enough?

Reviewer 2 ·

Basic reporting

I agreed with the authors that it is not possible to set a threshold age of mature forest at global scale based on biomass. Nevertheless, it is still unclear to me how the authors want to prove in this paper. I would suggest that authors define what is meant by "mature forest" to their understanding before discussing any other issue related to defining the threshold age of mature forest.

It is unclear what is the implication or consequence of setting or not setting a threshold age of mature forest globally? Does it tell anything by setting or not setting such threshold age of mature forest?

Forest biomass and forest carbon biomass have different meaning and units. Authors may refer to the terms precisely in order to avoid confusion.

In forestry practice at local scale or even regional scale, two important terms are usually used to describe forest biomass increment, growth, and forest maturity. They are Mean Annual Increment and Current Annual Increment or Periodic Annual Increment (MAI, CAI, PAI). Authors may want to extend their discussions to include why MAI, CAI, PAI were not part of the discussions when it comes to defining forest maturity.

Experimental design

No Comments

Validity of the findings

Results from the models should have been compared with observations or results from well-recognized models.

Additional comments

L233: Would it be "Problems with setting a threshold age of mature tree"?

---

## Round 0.2 · accepted · Accept

I am satisfied that the authors have now addressed the minor issues in the previous version.